# LC-MS/MS-Based Metabolomic Profiling of Constituents from *Glochidion velutinum* and Its Activity against Cancer Cell Lines

**DOI:** 10.3390/molecules27249012

**Published:** 2022-12-17

**Authors:** Syed Luqman Shah, Kashif Bashir, Hafiz Majid Rasheed, Jamil Ur Rahman, Muhammad Ikram, Abdul Jabbar Shah, Kamlah Ali Majrashi, Sulaiman Mohammed Alnasser, Farid Menaa, Taous Khan

**Affiliations:** 1Department of Pharmacy, COMSATS University Islamabad, Abbottabad Campus, Abbottabad 22060, Pakistan; 2Department of Pharmaceutical Sciences, Pak-Austria Fachhochschule, Institute of Applied Sciences and Technology, Mang, Khanpur Road, Haripur 22650, Pakistan; 3Biological Sciences Department, College of Science and Arts, King Abdulaziz University, Rabigh 21911, Saudi Arabia; 4Department of Pharmacology and Toxicology, Unaizah College of Pharmacy, Qassim University, Buraydah 52571, Saudi Arabia; 5Department of Oncology and Nanomedicine, California Innovations Corporation, San Diego, CA 92037, USA

**Keywords:** breast cancer, chemical profiling, Global Natural Product Social (GNPS) Molecular Networking, *Glochidion velutinum*, LC-MS/MS, MTT assay, prostate cancer

## Abstract

This study aimed to establish the phytochemical profile of *Glochidion velutinum* and its cytotoxic activity against prostate cancer (PC-3) and breast cancer (MCF-7) cell lines. The phytochemical composition of *G. velutinum* leaf extract and its fractions was established with the help of total phenolic and flavonoid contents and LC-MS/MS-based metabolomics analysis. The crude methanolic extract and its fractions were studied for pharmacological activity against PC-3 and MCF-7 cell lines using the MTT assay. The total phenolic content of the crude extract and its fractions ranged from 44 to 859 µg GAE/mg of sample whereas total flavonoid contents ranged from 20 to 315 µg QE/mg of sample. A total of forty-eight compounds were tentatively dereplicated in the extract and its fractions. These phytochemicals included benzoic acid derivatives, flavans, flavones, *O*-methylated flavonoids, flavonoid *O*- and *C*-glycosides, pyranocoumarins, hydrolysable tannins, carbohydrate conjugates, fatty acids, coumarin glycosides, monoterpenoids, diterpenoids, and terpene glycosides. The crude extract (IC_50_ = 89 µg/mL), the chloroform fraction (IC_50_ = 27 µg/mL), and the water fraction (IC_50_ = 36 µg/mL) were found to be active against the PC-3 cell line. However, the crude extract (IC_50_ = 431 µg/mL), the chloroform fraction (IC_50_ = 222 µg/mL), and the ethyl acetate fraction (IC_50_ = 226 µg/mL) have shown prominent activity against breast cancer cells. Moreover, *G. velutinum* extract and its fractions presented negligible toxicity to normal macrophages at the maximum tested dose (600 µg/mL). Among the compounds identified through LC-MS/MS-based metabolomics analysis, epigallocatechin gallate, ellagic acid, isovitexin, and rutin were reported to have anticancer activity against both prostate and breast cancer cell lines and might be responsible for the cytotoxic activities of *G. velutinum* extract and its bioactive fractions.

## 1. Introduction

After cardiovascular diseases, cancer is the major cause of death [1]. Though the modern age has advanced pharmaceuticals for cancer treatment, we are still deprived of the radical cure and patients often undergo miserable adverse effects while receiving treatments [2]. Cancer treatments include surgical resection of the cancerous mass, radiotherapy, and chemotherapy [3]. Common problems with the available therapeutic options are a high risk of adverse reactions, resistance, ineffectiveness, and high cost. Cancer diagnoses are constantly increasing day by day, and it is expected that the incidence will increase in the future. Based on incidence rates, female breast cancer is the most commonly diagnosed cancer followed by lung, colorectal, prostate, and stomach cancer. Based on mortality, lung cancer is at the top, followed by colorectal, liver, stomach, and breast cancer. The global cancer burden is expected to be about 28.4 million cases in 2040 [4].

Prostate cancer is considered one of the most frequently occurring cancers in the male population [5,6]. The accumulation of somatic mutations in the prostate epithelial cell genome during a patient’s lifetime is thought to be a significant link between prostate cancer and the disease. Genes that control cell growth, cell proliferation, and cell death are the most common targets for mutations [5]. Metastasis to other organs results in more complications and death. Resistance of the tumor cells is also a major problem in the treatment of prostate cancer while treating it with cytotoxic agents. Docetaxel was introduced and approved by the FDA in 2004 for the treatment of prostate cancer, but later on, resistance was developed against docetaxel. Other new chemical entities were also approved by the FDA, but questions were raised regarding their safety, efficacy, and cost. During the past decade, treatment strategies for patients with advanced prostate cancer relating to various stages after treatment with curative intent, as well as castration-resistant prostate cancer, have extensively evolved with the introduction and approval of several new agents including sipuleucel-T, radium-223, abiraterone, enzalutamide, and cabazitaxel [7]. 

Breast cancer is the most commonly diagnosed cancer, worldwide. The prevalence of breast cancer has increased over the past several decades [4]. There are 2.3 million new cases of breast cancer in both sexes combined today [4]. Breast cancer typically begins as ductal hyper-proliferation and progresses to benign tumors or even metastatic carcinomas when they are repeatedly stimulated by numerous carcinogenic stimuli [8]. Both oncogene and anti-oncogene mutations and abnormal amplification play critical roles in the development and progression of breast cancer [8]. The combination of pertuzumab, trastuzumab, and docetaxel has been approved for the treatment of breast cancer [8]. All of these agents have shown a significant improvement in overall survival, but these are associated with severe side effects such as non-selectivity, resistance, low bioavailability, and efficacy [9]. Such issues reduce the motivation of researchers focusing to treat cancer patients [7]. 

Traditionally, plants have been used in all communities for healing different diseases, and experimental results highlight the potential of plants as sources of anticancer compounds [10,11]. In several cases, phytochemical constituents have been used directly or chemically modified in the development of anticancer medications. More than 60% of drugs used for the treatment of cancer are derived from natural resources, according to the Food and Drug Administration (FDA) [7]. 

*Glochidion velutinum* Wight belongs to the family Euphorbiaceae [12]. It is mostly found in Pakistan, Burma, Nepal, China, Vietnam, and India. It is commonly known as Mattachar, Kaalikaath, Velvety Melon Feather foil, and Downy Melon Feather foil [13,14]. It has traditionally been used to treat cancer, diabetes, inflammation, wounds, coughs, and diarrhea. Flavonoids, glycosides, saponins, and alkaloids are known to be the constituents of *G. velutinum*. The reported pharmacological activities of *G. velutinum* include antibacterial, antioxidant, cytotoxic, antidiabetic, anti-inflammatory, and antiurolithiatic activity. *G. velutinum* has been traditionally used in cancer treatment [15]; however, according to the literature, its extract, fractions, and constituents have not been pharmacologically evaluated for anticancer potential against various cell lines including prostate cancer. Furthermore, it has not been investigated for detailed metabolite profiling, isolation, and characterization of its various constituents. Therefore, the current study focused on the phytochemical profiling of *G. velutinum* extract and its fractions, and their evaluation against prostate (PC-3) and breast cancer (MCF-7) cell lines. 

## 2. Results and Discussion

### 2.1. Phytochemical Analysis

#### 2.1.1. Total Phenolic Contents

Based on the results obtained, the crude extract and fractions of *G. velutinum* showed a significant amount of total phenolic content, as shown in Table 1. The highest phenolic content was shown by the ethyl acetate fraction (859 ± 1.3 µg GAE/mg) followed by crude extract (588 ± 3 µg GAE/mg), chloroform (266 ± 1 µg GAE/mg), and aqueous (77 ± 2.3 µg GAE/mg) fractions. Polyphenols are well known for their anti-inflammatory, antioxidant, and anticancer properties [16]. In this study, a significant amount of phenolic content was estimated in the crude extract and fractions of *G. velutinum*. Therefore, the presence of a remarkable number of polyphenols depicts the pharmacological value of *G. velutinum*.

#### 2.1.2. Total Flavonoid Contents

The obtained results from total flavonoid contents (TFC) showed that the extract and its fractions contain a considerable quantity of TFCs, as shown in Table 2. The ethyl acetate fraction showed the maximum quantity of flavonoid contents (315 ± 1 µg QE/mg) followed by crude extract (118 ± 2 µg QE/mg) and the chloroform (79 ± 1.3 µg QE/mg) fraction. Flavonoids are plant bioactive constituents of great interest due to their remarkable antioxidant, anti-inflammatory, antibacterial, antifungal, and anticancer properties [17]. Hence, the presence of a remarkable number of flavonoids in the crude extract and its fractions also signifies the importance of *G. velutinum* for medicinal use, especially in inflammation and cancer.

#### 2.1.3. LC-MS/MS-Based Chemical Profiling

Based on the anti-prostate and anti-breast cancer activities of the crude extract of *G. velutinum* leaves, it was investigated to establish the detailed phytochemical profile, using the LC-MS/MS (tandem mass) and GNPS-based metabolomics platform. Further, the fraction-level biological activity against the prostate cancer cell line (PC-3) and the breast cancer cell line (MCF-7) was determined for the polarity-based fractions obtained through solvent–solvent extraction from the *G. velutinum* crude extract. The fractions were also analyzed through tandem mass spectrometry and the GNPS molecular networking platform. Based on previous literature, none of the species from the genus Glochidion were studied for detailed phytochemical profiling using LC-MS/MS-based modern metabolomics approaches. 

HPLC-MS^n^ is routinely employed in the separation and tentative identification of complex mixtures of natural product origin [18,19]. Therefore, this technique was used to carry out a comprehensive and detailed phytochemical analysis of the crude extract and fractions of *G. velutinum* leaves. A total of 46 compounds were tentatively identified in the crude extract and subsequent polarity-based fractions of *G. velutinum* leaves. Based on the phytochemical analysis results, the crude extract was found to contain flavonoid glycosides, coumarins, fatty acids, and sugars. The chloroform fraction was found to be most bioactive against prostate cancer and breast cancer. It was found to contain flavonoid glycosides, flavans, pyranocoumarins, hydrolysable tannins, monoterpenoids, carbohydrate conjugates, and fatty acids, as shown in Figure 1 and Figure 2. Ethyl acetate, being another bioactive fraction against both types of cancer, was found to contain benzoic acid derivatives, coumarin glycosides, flavans, flavones, flavonoid glycosides, diterpenoids, terpene glycosides, pyranocoumarins, O-methylated flavonoids, and linoleic acid derivatives, as shown in Figure 3. The various constituents observed in the crude extract and fractions from *G. velutinum* leaves are listed in Table 3 along with their *m*/*z* values in negative ion mode, MS/MS fragmentation patterns, and retention time.

The reported chemical constituents that were isolated from the plant include D-mannitol, β-amyrin, stigmasterol, glochidonol, glochidone, glochidiol, betulin, β-daucosterol, glochidioside, glochidioside N, glochidioside Q, and epimachaerinic acid [11,20]. The major phenolic compounds identified in the *G. velutinum* extract and its fractions include ellagic acid (**9**), gallic acid (**32**), 2-hydroxycinnamic acid (HCA) (**34**), epigallocatechin gallate (EGCG) (**26**), gallocatechol (**3**), trans-piceid (**24**), and 1,3,6-tri-O-galloylglucose (**37**). The flavonoid glycosides include isoquercitrin (**28**), rutin (**10**), rhoifolin (**41**), kaempferol 3-O-glucoside (**39**), homoorientin (**38**), vitexin-2-O-rhamnoside (**27**), vicenin-2 (**25**), isovitexin (**5**) and myricetin 3-galactopyranoside (**6**), hyperoside (**7**), and luteolin 4’-O-glucoside, (**42**) whereas isokaempferide (**35**), 5,6,2′-trimethoxy flavone (**15**), catechin (**33**), and luteolin (**40**) were identified as simple flavonoids. Trehalose (**2**), arabinose (**44**), and maltotriose (**29**) were observed as sugars. Quinic acid (**1**) was identified as a cyclic polyol. Digalactosylmonoglycerol (DGMG) (**19**) and monogalactosylmonoglycerol (MGMG) (**31**) were identified as glycosyl glycerol. 9-Hydroxy-10E,12Z,15Z-octadecatrienoic acid (**20**), FA 18:4+2O (**18**), FA 18:1+3O (**12**), stearidonate (18:4(n−3)) (**13**), methyl (2E,4E,8E)-7,13-dihydroxy-4,8,12-trimethyltetradeca-2,4,8-trienoate (**16**), and (10E,15E)-9,12,13-trihydroxyoctadeca-10,15-dienoic acid (**11**) were identified as fatty acids (Table 3 and Table 4). Roseoside (**36**) was identified as a fatty acyl glycoside.

Some typical fragmentation patterns were distinctive for the tentative identification of flavonoid glycoside showing galactose (162), rhamnose (146), and glucose (162) for O-glycosides [21]. Moieties such as 60, 90, and 120 were characteristic of C-glycosides [22]. Gallic acid and HCA fragmented with the loss of CO_2_ moiety [23].

#### 2.1.4. Anticancer Activity

In this study, the anticancer potential of the crude extract and fractions of *G. velutinum* was evaluated using the MTT assay. The treatment of PC-3 cells with *G. velutinum* extract and its fractions for 24 h resulted in the reduction of cell viability in comparison to DMSO (control) treated cells (no cell death). From Figure 4, it can be observed that the n-hexane fraction presented the least cytotoxic effect (IC_50_ = 325 µg/mL) on PC-3 cells, whereas ethyl acetate and n-butanol fractions showed a moderate effect with respective IC_50_ values of 196 µg/mL and 123 µg/mL, respectively. The stronger effects were observed for the chloroform fraction (IC_50_ = 27 µg/mL), followed by the aqueous fraction (IC_50_ = 36 µg/mL) and *G. velutinum* extract (IC_50_ = 89 µg/mL).

Similarly, the treatment of MCF-7 cells with *G. velutinum* extract and its fractions showed a decrease in cell viability in comparison to DMSO-treated cells. From Figure 5, among the tested samples, the n-butanol fraction presented the least cytotoxic effect (24% cell growth inhibition) on MCF-7 cells, whereas the aqueous and n-hexane fractions showed a moderate effect with IC_50_ values of 522 µg/mL and 523 µg/mL, respectively. The stronger effects were observed for the chloroform fraction (IC_50_ = 222 µg/mL), followed by the ethyl acetate fraction (IC_50_ = 226 µg/mL) and *G. velutinum* extract (IC_50_ = 431 µg/mL). The IC_50_ values of *G. velutinum* extract and its fractions for the MCF-7 and PC-3 cell lines have also been given in Table 5.

Plant-derived natural products remain the major contributors to pharmacotherapy, especially for cancer and infectious diseases [24]. Among these phytoconstituents, polyphenols have been extensively explored for the treatment of cancer [25,26,27]. In this work, phenolics, flavonoids, fatty acids, terpenoids, coumarins, and sugars were identified as major constituents of *G. velutinum* extract and its fractions. Based on literature surveys of the identified compounds in the crude extract and its fractions, epicatechin gallates were previously reported for anticancer activity for prostate cancer cell lines [28]. Epigallocatechin-3-gallate (EGCG) inhibits PC-3 prostate cancer cell proliferation via MEK-independent ERK1/2 activation [29]. Isovitexin, ellagic acid, trehalose, and rutin were also identified as possessing activity against prostate cancer cell lines [30]. On the other hand, it is also noteworthy that the above-mentioned constituents, i.e., epigallocatechin-3-gallate (EGCG), isovitexin, isokaempferide, quinic acid, rhoifolin gallic acid, luteolin, ellagic acid, and rutin have also been shown to possess activity against breast cancer cell lines [31,32,33,34,35]. Hence, the presence of all these compounds may be responsible for the activity of the crude extract and its fractions against prostate and breast cancer cell lines. These results indicate that the fractions of *G. velutinum* extract may also be useful for the isolation of anticancer constituents, especially against prostate and breast cancer.

Moreover, the *G. velutinum* extract and its fractions were tested against peritoneal macrophages (at a maximum tested concentration of 600 µg/mL), and it was observed that the *G. velutinum* extract and its fractions induced negligible toxicity to normal macrophages in comparison to the control that is presented in Figure 6. Safety is the major concern for the development of novel therapeutic agents [9]. In this study, the low cytotoxic effect of the *G. velutinum* extract and its fractions against normal macrophages demonstrated a good safety margin of the tested samples. Based on earlier reports [9,36], selectively targeting cancer cells with minimum toxicity to normal cells prevents damaging effects on body organs [9]. Therefore, *G. velutinum* extract can be considered safe for the isolation of anticancer constituents.

## 3. Materials and Methods

### 3.1. Collection and Identification of the Selected Plant

The leaves of *G. velutinum* Wight were collected from Batrasi Reserved Forest, District Mansehra, Pakistan, in June. The plant specimen was authenticated by Dr. Abdul Majid, Assistant Professor, Department of Botany, Hazara University, Pakistan. The voucher specimen (GV-ZH-02/18) was deposited to the Department of Environmental Sciences Herbarium, COMSATS University Islamabad, Abbottabad Campus for future records.

### 3.2. Processing of Plant Material and Extraction

The collected plant material was shade-dried at ambient temperature (24–26 °C). The dried material was powdered (5 Kg) and subjected to extraction using methanol (15 L) with occasional shaking. The extraction was performed for 14, 7, and 3 days, respectively to completely exhaust the plant material. The extracted material was filtered through muslin cloth followed by Whatman filter paper. A vacuum rotary evaporator (Büchi Rotavapor^®^ R-300 Flawil, Switzerland) was used for the concentration of filtrates to get the crude extract [37]. The yield of *G. velutinum* crude extract was 410 g (8.2%).

### 3.3. Fractionation of Crude Extract

The crude extract (340 g) was suspended in distilled water and extracted with various organic solvents (*n*-hexane, chloroform, ethyl acetate, and *n*-butanol) in increasing order of polarity using established protocols [38,39]. The organic layers and final residual aqueous layer after solvent–solvent extraction were dried with the help of a rotary evaporator to obtain crude fractions. The crude extract of *G. velutinum* yielded various fractions including *n*-hexane (20 g), chloroform (180 g), ethyl acetate (15 g), *n*-butanol (35 g), and an aqueous fraction (40 g).

### 3.4. Phytochemical Evaluation

#### 3.4.1. Total Phenolic Contents (TPC)

The TPC of the *G. velutinum* extract and its fractions was determined using the Folin–Ciocalteu reagent (FCR) method according to the procedure reported by John et al., 2014 [40]. Gallic acid was used as the standard polyphenolic compound to construct a calibration curve for the measurement of the contents of the samples. The samples of gallic acid, plant extract, and fractions were prepared in methanol (0.5 mL) and mixed with FCR (1.5 mL). The mixture was incubated at room temperature for 5 min. Then, 4 mL of Na_2_CO_3_ solution (7.5%) was added to the above mixture and the volume was made up to 25 mL with distilled water. After a 30 min incubation, absorbance was measured at 765 nm with a UV/VIS spectrophotometer (Model UVD-3000, Labomed, Inc., Los Angeles, CA, USA) against distilled water as a blank. Results were expressed as micrograms of gallic acid equivalents (GAE)/mg of dry extract.

#### 3.4.2. Total Flavonoid Contents (TFC)

The TFC of the *G. velutinum* extract and its fractions was determined using the aluminum chloride colorimetric method according to the procedure reported by Vyas et al. (2015) with slight alteration [41]. The calibration curve for the determination of contents in the samples was constructed using quercetin as the standard compound. The samples of quercetin, plant extract, and fractions were dissolved in 0.5 mL methanol. In test tubes, the measured volumes (500 µL) of quercetin, crude extract, and fractions solution were placed. Each test tube was filled with distilled water (3 mL) and 0.3 mL sodium nitrite solution. After 5 min, 0.3 mL of aluminum chloride (10% *w*/*v*) and sodium hydroxide (1 M) solutions were added to the test tubes. Finally, distilled water was used to adjust the volume (up to 10 mL). The absorbance was measured at 415 nm using a UV-visible spectrophotometer (Model UVD-3000, Labomed, Inc., Los Angeles, CA, USA). The calibration curve for quercetin was generated by plotting the absorbance against different concentrations of the extract and its fractions. The results were expressed as micrograms of quercetin equivalents (QE)/mg of dry extract.

#### 3.4.3. LC-MS/MS-based Metabolomic Profiling

The LC-MS/MS analysis of the extract and various fractions was performed according to the procedure mentioned in Bashir et al. (2021) with modifications where required using negative ion mode [21]. Briefly, samples of methanolic extract of *G. velutinum* and fractions were prepared by dissolving in HPLC grade methanol at a concentration of 1 mg/mL. The samples were vortexed, sonicated, and then filtered through a 0.45 µm membrane filter and transferred to Orbitrap HPLC vials. The Dionex Ultimate 3000 UHPLC system coupled with Velos Pro Orbitrap Mass spectrometer was employed. A reverse-phase (C-18) analytical column (50 mm, 2.1 mm, 1.6 µm) was used as the stationary phase while acetonitrile and water both acidified with 0.1% formic acid was employed as the mobile phase in the gradient elution HPLC program. An amount of 10 µL of each sample was injected into the column and a flow rate of 0.5 mL/min was maintained. The LC-MS/MS analysis was carried out using the electrospray ionization (ESI) technique in negative ion mode. The Global Natural Product Social (GNPS) molecular networking platform (https://gnps.ucsd.edu/ProteoSAFe/static/gnps-splash.jsp, accessed on 30 October 2022) and its various tools were used for the dereplication and tentative identification of the various phytochemical constituents of the crude extract and its fractions [22].

### 3.5. Evaluation of the Anticancer Activity

The MTT assay was performed to evaluate the anticancer properties of the extract and its fractions [9,42].

#### 3.5.1. Cell Culture

For cancer cells, cell lines (PC-3, and MCF-7) were purchased from American Type Culture Collection (ATCC) and grown in a Dulbecco’s Modified Eagles Medium (DMEM) with the addition of 10% fetal bovine serum (FBS), L-glutamine, and antibiotics. The cells were kept at 37 °C in a humified environment of 5% CO_2_ and 95% oxygen. Cancer cells were cultured in culture flasks at specific concentrations, such as 1 × 10^5^ in 10 mL of complete culture media for a 25 cm^2^ flask. From these culture flasks, cells were used for further analysis.

For normal cells, macrophages were isolated from the peritoneum of mice. The cells were washed and seeded in cell culture flasks of 25 cm^2^ supplemented with Dulbecco’s Modified Eagle Medium (DMEM) media with 10% fetal bovine serum (FBS) and 1% penicillin/streptomycin antibiotics. Then, culture flask cells were seeded in a 96-well plate and were treated with the crude extract and fractions of *G. velutinum* at a concentration of 600 µg/mL [43].

#### 3.5.2. MTT Assay

For the MTT assay, the cultured cells were trypsinized and plated individually in 96-well plates (Costar^®^, Corning Inc., Corning, NY, USA) at 10,000-per-well density. After 24 h, cells were incubated with various concentrations (75–600 µg/mL) of the crude extract and its fractions for 24 h. Similarly, DMSO, a control (<1%), was mixed with media and added to the control wells. Then, the media was aspirated, and fresh media (100 µL) containing the MTT reagent (BioShop Inc., Burlington, ON, Canada) was added to each well. Plates were incubated for an additional 4 h. For the solubilization of formazan crystals, 100 µL of dimethyl sulfoxide (DMSO) was added to each well. The plates were scanned at 492 nm using a plate reader (Model VEGA 500, Easy Access International Co., Ltd., Shanghai, China). Experiments were conducted in triplicate and percent cell viability was calculated by the following formula:$$\%\;\mathrm{Viability}=\frac{\mathrm{Absorbance}\;\mathrm{of}\;\mathrm{sample}}{\mathrm{Absorbance}\;\mathrm{of}\;\mathrm{negative}\;\mathrm{control}}\times 100$$

### 3.6. Statistical Analysis

The results were statistically analyzed through a two-tailed Student’s *t*-test by applying GraphPad Prism (version 5.0). The data were presented as MEAN ± standard deviation with a confidence interval of 95%.

## 4. Conclusions

In conclusion, *G. velutinum* extract and its fractions have a remarkable number of polyphenols. Moreover, the present work has established the detailed phytochemical profile of *G. velutinum*, which indicated 46 compounds in the crude extract and its fractions comprising mostly phenolics, flavonoids, fatty acids, terpenoids, coumarins, and sugars. In addition, *G. velutinum* extract and its fractions presented promising cytotoxic effects against prostate and breast cancer cells. Furthermore, the traditional use of this plant as an anticancer treatment was confirmed and strengthened by these results. This study suggested that *G. velutinum* has anticancer potential and may be used for the isolation and development of relatively safer anticancer drugs.

## Figures and Tables

**Figure 1 molecules-27-09012-f001:**
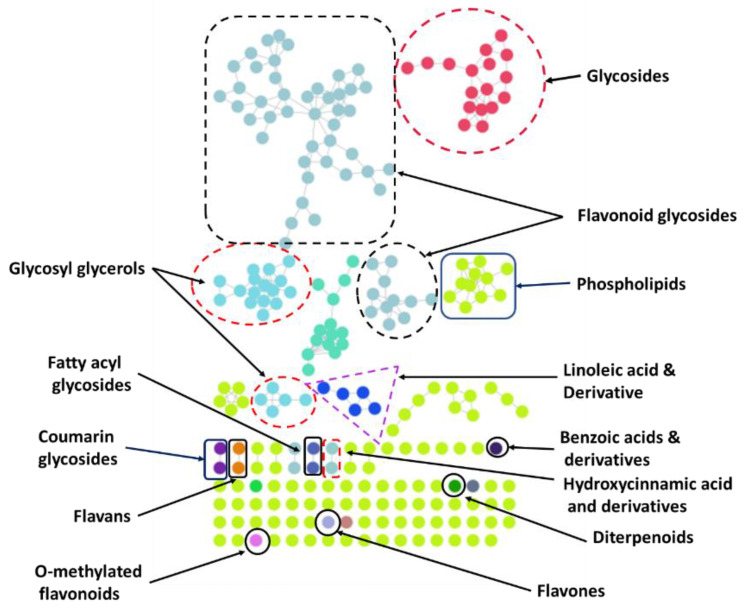
Phytochemical classes of bioactive chloroform fraction from *G. velutinum* leaves using LC-MS/MS-based molecular networking analysis.

**Figure 2 molecules-27-09012-f002:**
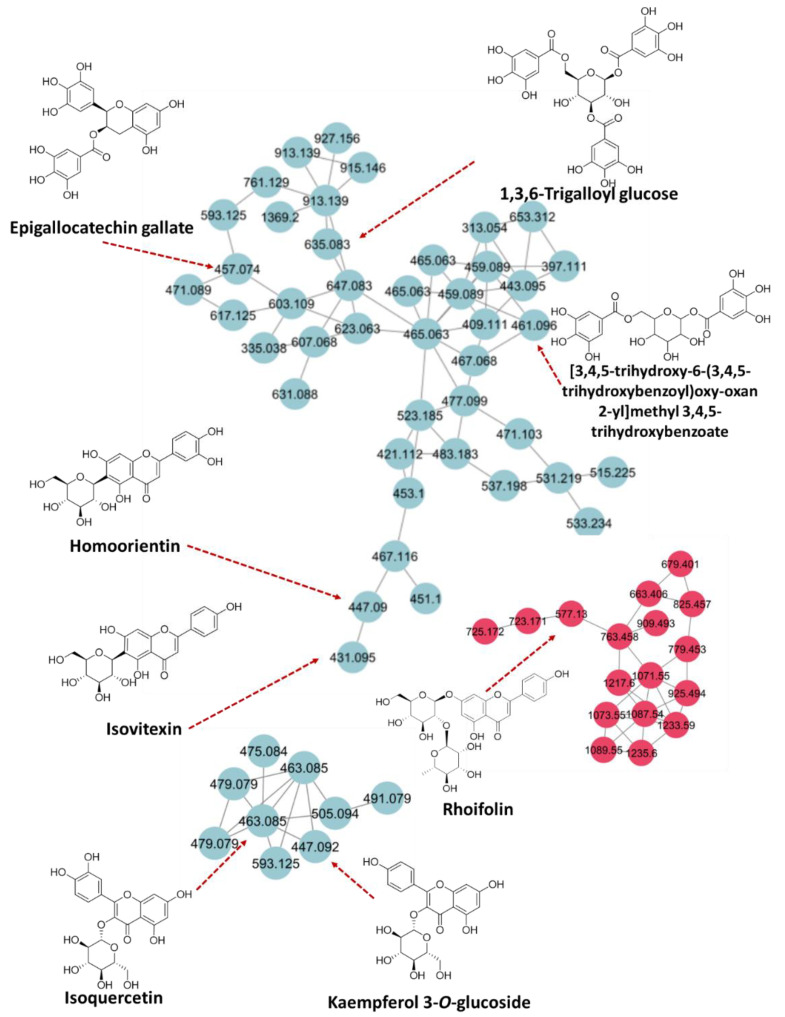
Chemical constituents observed in the chloroform fraction of *G. velutinum* leaves using LC-MS/MS and GNPS classical molecular networking.

**Figure 3 molecules-27-09012-f003:**
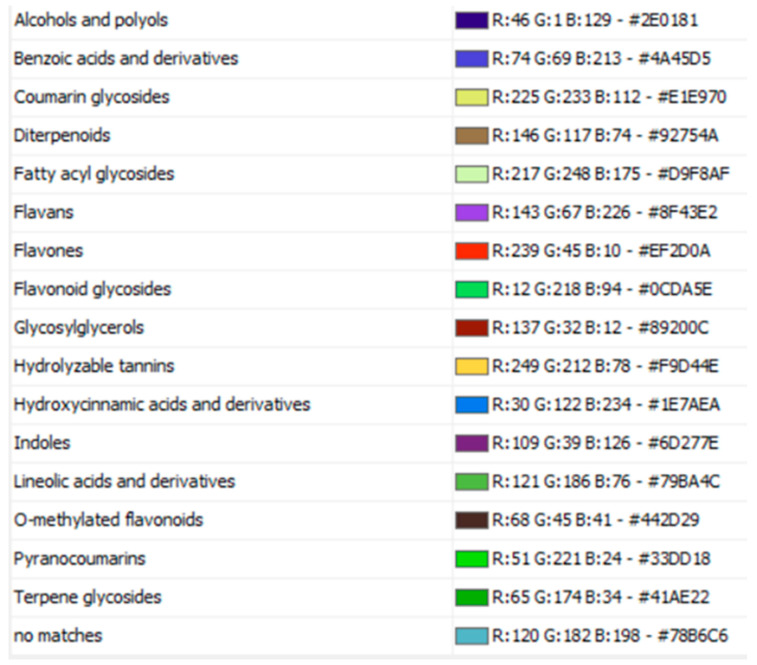
Classes of chemical constituents were identified using GNPS molecular networking and MolnetEnhancer technique to identify the chemical space present within ethyl acetate fraction of *G. velutinum* leaves.

**Figure 4 molecules-27-09012-f004:**
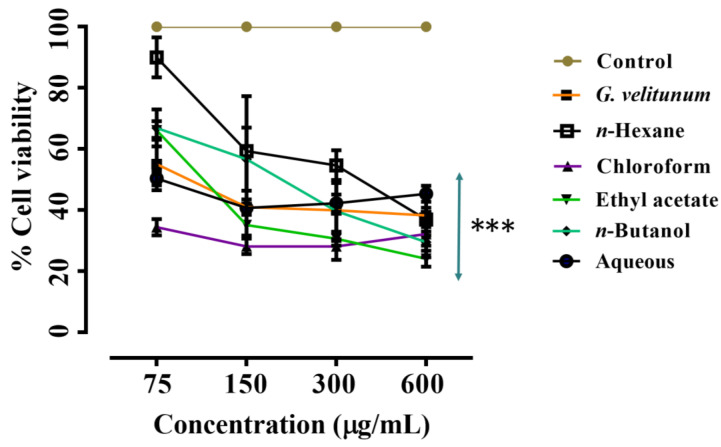
Presentation of growth inhibitory effects of *G. velutinum* extract and its fractions on PC-3 cells. The results are expressed as a percentage of the negative control. Data are means ± SD of three independent experiments. *** represent *p* ≤ 0.001.

**Figure 5 molecules-27-09012-f005:**
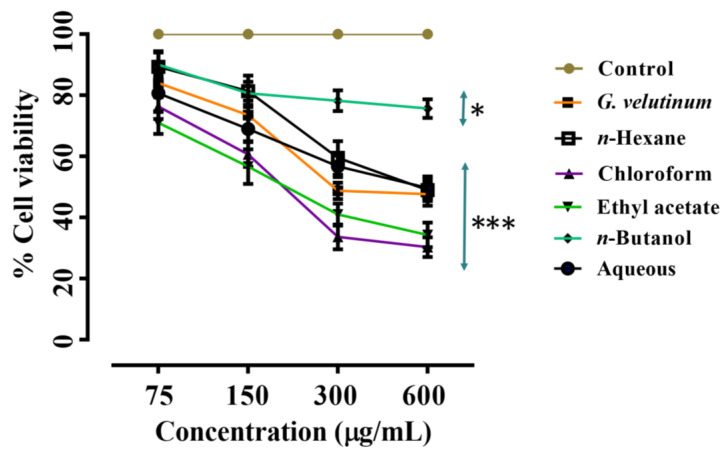
Presentation of growth inhibitory effects of *G. velutinum* extract and its fractions on MCF-7 cells. The results are expressed as a percentage of the negative control. Data are means ± SD of three independent experiments. * and *** represent *p* ≤ 0.05 and *p* ≤ 0.001, respectively.

**Figure 6 molecules-27-09012-f006:**
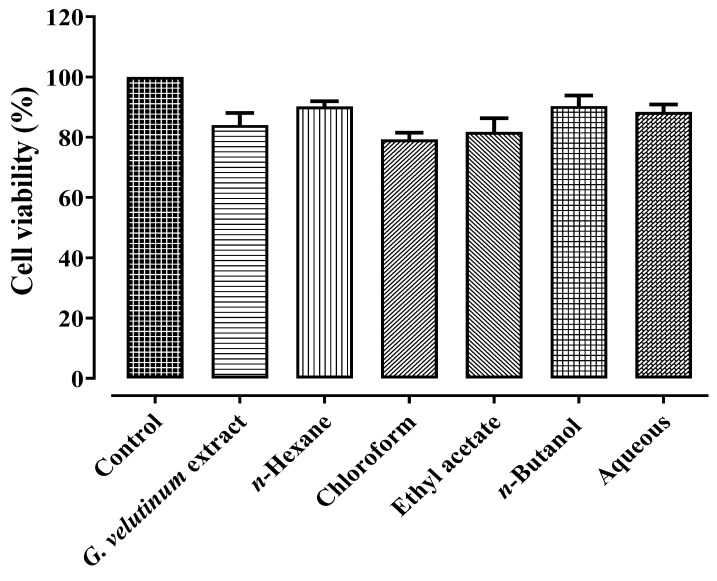
Presentation of growth inhibitory effects of *G. velutinum* extract and its fractions (at maximum tested concentration) against normal cells.

**Table 1 molecules-27-09012-t001:** Total phenolic contents of crude extract and fractions of *G. velutinum*.

Entry No.	Extract	Total Phenol (µg GAE/mg)
1	Crude	588 ± 3.0
2	*n*-Hexane	54 ± 1.2
3	Chloroform	266 ± 1.0
4	Ethyl acetate	859 ± 1.0
5	*n*-Butanol	44 ± 2.1
6	Aqueous	77 ± 2.3

**Table 2 molecules-27-09012-t002:** Total flavonoid contents of crude extract and fractions of *G. velutinum*.

Entry No.	Extract	Total Flavonoid (µg QE/mg)
1	Crude	118 ± 2.0
2	*n*-Hexane	20 ± 1.1
3	Chloroform	79 ± 1.3
4	Ethyl acetate	315 ± 1.0
5	*n*-Butanol	24 ± 1.5
6	Aqueous	29 ± 1.4

**Table 3 molecules-27-09012-t003:** The phytochemical profile of *G. velutinum* leaves was established with the help of LC-MS/MS-based molecular networking.

Crude Extract of *G. velutinum*						
Entry #	Compound Name	RT	*m*/*z*	MS^2^ Fragmentation Pattern	Molecular Formula	Exact Mass
1	Quinic acid	0.43	191.0568	111.1887 (100), 173.3876, 85.1423, 127.066, 146.063, 93.184	C_7_H_12_O_6_	192.0633
2	Trehalose	0.55	341.1099	179.3955 (100), 161.3627, 143.3026, 119.2360, 113.2109	C_12_H_22_O_11_	342.1162
3	Gallocatechol	1.14	305.0642	221.1006 (100), 179.1527, 273.0852, 261.1541	C_15_H_14_O_7_	306.0739
4	Citric acid	1.71	191.0566	147.3730 (100), 111.1887, 85.1423	C_6_H_8_O_7_	192.0270
5	Isovitexin	2.96	431.0991	311.1170 (100), 341.1700, 269.1172, 367.1852	C_21_H_20_O_10_	432.1056
6	Myricetin, 3-Galactopyranoside	3.71	479.1910	316.6702 (100), 193.4596, 271.5861	C_21_H_20_O_13_	480.0903
7	Hyperoside	4.02	463.5582	301.5882 (100)	C_21_H_20_O_12_	464.0954
8	CGHTM, ME	4.05	447.0950	285.5274 (100), 327.6831, 313.7282	C_17_H_26_O_11_	406.1475
9	Ellagic acid	6.07	300.9967	257.0632 (100), 229.0459, 185.1000, 271.0589, 151.1032	C_14_H_6_O_8_	302.0062
10	Rutin	6.38	609.0831	463.1980 (100), 301.1688	C_27_H_30_O_16_	610.1533
11	9,12,13,TriHODE	6.76	327.2881	229.2324 (100), 291.2702, 211.2155, 171.2333	C_18_H_32_O_5_	328.2249
12	FA 18:1+3O	6.84	329.1445	229.2349 (100), 171.1871, 211.1918, 293.2625	C_18_H_34_O_5_	330.2406
13	Stearidonate (18:4n3)	7.35	277.206	233.6909 (100), 205.7536, 179.5107	C_18_H_27_O_2_^-^	275.2011
14	8,8-DDPCM	8.19	339.8762	183.4248 (100), 299.6149, 197.4348	C_19_H_18_O_6_	342.1103
15	5,6,2′-Trimethoxyflavone	8.77	311.166	183.1256 (100), 247.3156, 198.1505	C_18_H_16_O_5_	312.0997
16	MDTT	9.06	311.1662	183.373 (100), 149.344, 271.601, 247.682	C_18_H_30_O_4_	310.2144
***n*-Hexane fraction**						
17	Azelaic acid	6.24	187.2562	125.1266 (100), 143.2794, 97.4454, 159.1357	C_9_H_16_O_4_	188.1048
12	FA 18:1+3O	6.84	329.1445	229.2349 (100), 171.1871, 211.1918, 293.2625	C_18_H_34_O_5_	330.2406
18	FA 18:4+2O	7.15	307.1800	97.0354 (100), 267.0800	C_18_H_28_O_4_	308.1987
19	DGMG 18:3	7.41	721.3603	675.5287 (100), 397.2563, 415.2704	C_33_H_56_O_14_	676.3670
20	9-HOA	7.73	275.2000	231.2642 (100), 177.2366, 203.3373, 255.2677, 239.4132	C_18_H_30_O_3_	294.2194
21	13-HODE	7.92	295.3241	195.2709 (100), 171.2408, 179.2919, 251.3765, 181.2437	C_18_H_32_O_3_	296.2351
22	Docosanol	8.97	325.2862	183.1292 (100)	C_22_H_46_O	326.3548
**Chloroform fraction**						
23	*p*-Mentha-1-ene-6-one	3.59	216.5700	171.0872 (100), 144.1471, 125.9763, 168.9540	C_10_H_16_O	152.1201
9	Ellagic acid	6.07	300.9967	257.0632 (100), 229.0459, 185.1000 271.0589, 151.1032	C_14_H_6_O_8_	302.0062
24	*trans*-piceid	5.70	433.2105	387.2457 (100), 225.1868, 193.112, 313.1065	C_20_H_22_O_8_	390.1314
25	Vicenin-2	5.70	593.2812	473.1857 (100), 353.1384, 503.181, 383.1578, 413.2072	C_27_H_30_O_15_	594.1584
26	EGCG	5.85	457.0732	331.1129 (100), 305.1384, 169.0689, 413.1637, 193.0952	C_22_H_18_O_11_	458.0849
27	Vitexin-2-*O*-rhamnoside	6.01	577.1510	311.1990 (100), 413.2145, 341.1857, 293.2155, 395.136, 283.16	C_27_H_30_O_14_	578.1635
28	Isoquercetin	6.01	463.3657	301.1005 (100), 316.0813, 343.0984	C_21_H_20_O_12_	464.0954
10	Rutin	6.38	609.0831	463.1980 (100), 301.1688	C_27_H_30_O_16_	610.1533
11	9,12,13,TriHODE	6.76	327.2881	229.2324 (100), 291.2702, 211.2155, 171.2333	C_18_H_32_O_5_	328.2249
29	Maltotriose	7.50	449.7303	502.4152 (100), 503.2143, 418.2113, 491.4700, 523.2600, 371.3013	C_18_H_32_O_16_	504.1690
30	9,10-DiHOME	7.52	313.2362	201.1504 (100), 183.1968, 293.0565, 277.2592, 171.1771, 195.1924	C_18_H_34_O_4_	314.2457
15	5,6,2′-Trimethoxyflavone	8.77	311.166	183.1256 (100), 247.3156, 198.1505	C_18_H_16_O_5_	312.0997
31	MGMG 18:3	7.72	559.308	331.4297 (100), 305.10051, 169.0092	C_27_H_46_O_9_	514.3141
19	DGMG 18:3	7.41	721.3603	675.5287 (100), 397.2563, 415.2704	C_33_H_56_O_14_	676.3670
14	8,8-DDPCM	8.19	339.8762	183.4248 (100), 299.6149, 197.4348	C_19_H_18_O_6_	342.1103
**Ethyl acetate fraction**						
32	Gallic acid	0.42	169.0149	125.2083 (100), 141.1750, 81.4432, 69.1584	C_7_H_6_O_5_	170.0215
3	Gallocatechol	1.14	305.0642	221.1006 (100), 179.1527, 273.0852, 261.1541	C_15_H_14_O_7_	306.0739
33	Catechin	2.20	289.1863	245.1277 (100), 205.1055, 179.1031	C_15_H_14_O_6_	290.0790
5	Isovitexin	2.96	431.0991	311.1170 (100), 341.1700, 269.1172, 367.1852	C_21_H_20_O_10_	432.1056
34	2-HCA	3.50	163.0392	119.0751(100), 135.1361	C_9_H_8_O_3_	164.0473
35	Isokaempferide	4.67	301.5982	257.5770 (100), 229.4959, 272.5718	C_16_H_12_O_6_	300.0633
36	Roseoside	5.38	387.2834	207.0929 (100), 163.1177, 225.0903	C_19_H_30_O_8_	386.1940
37	1,3,6-tri-*O*-galloylglucose	5.80	635.0832	465.1552 (100), 483.1615, 313.1343, 423.1522, 591.1775, 221.1012	C_27_H_24_O_18_	636.0962
26	EGCG	5.85	457.0732	331.1129 (100), 305.1384, 169.0689, 413.1637, 193.0952	C_22_H_18_O_11_	458.0849
38	Homoorientin	5.89	447.0904	327.0962 (100), 357.0830, 285.0861, 313.0800	C_21_H_20_O_11_	448.1005
28	Isoquercetin	6.01	463.3657	301.1005 (100), 316.0813, 343.0984	C_21_H_20_O_12_	464.0954
39	Kaempferol 3-*O*-glucoside	6.13	447.0922	285.0800 (100), 301.0750, 307.1052 327.1223	C_21_H_20_O_11_	448.1005
40	Luteolin	6.59	285.0380	241.1324 (100), 175.0822, 199.1310, 217.0913, 151.0380	C_15_H_10_O_6_	286.0477
41	Rhoifolin	6.63	577.1302	269.1225 (100), 431.1875, 413.1603, 307.1234, 327.1476	C_27_H_30_O_14_	578.1635
11	9,12,13-TriHODE	6.76	327.2881	229.2324 (100), 291.2702, 211.2155, 171.2333	C_18_H_32_O_5_	328.2249
12	FA 18:1+3O	6.84	329.1445	229.2349 (100), 171.1871, 211.1918, 293.2625	C_18_H_34_O_5_	330.2406
22	Docosanol	8.97	325.2862	183.1292 (100)	C_22_H_46_O	326.3548
14	8,8-DDPCM	8.19	339.8762	183.4248 (100), 299.6149, 197.4348	C_19_H_18_O_6_	342.1103
***n*-Butanol fraction**						
1	Quinic acid	0.43	191.057	111.1887 (100), 85.1423, 127.2440, 146.063, 93.184	C_7_H_12_O_6_	192.0633
2	Trehalose	0.50	341.1099	179.3955 (100), 161.3627, 143.3026, 119.2360, 113.2109	C_12_H_22_O_11_	342.1162
5	Isovitexin	2.96	431.0991	311.1170 (100), 341.1700, 269.1172, 367.1852	C_21_H_20_O_10_	432.1056
7	Hyperoside	4.02	463.0932	301.5882 (100)	C_21_H_20_O_12_	464.0954
42	Luteolin 4’-*O*-glucoside	4.16	447.0980	284.582 (100), 327.7294, 315.7570, 255.572	C_21_H_20_O_11_	448.1005
**Aqueous fraction**						
2	Trehalose	0.55	341.1099	179.3955 (100), 161.3627, 143.3026, 119.2360, 113.2109	C_12_H_22_O_11_	342.1162
43	6,7-Dimethyl-4-hydroxycoumarin	0.67	188.3989	131.3810 (100), 147.4211, 85.2340	C_11_H_10_O_3_	190.0629
4	Citric acid	1.71	191.0566	147.3730 (100), 111.1887, 85.1423	C_6_H_8_O_7_	192.0270
44	Arabinose	3.36	174.9568	131.2210 (100), 147.2940, 155.4096	C_5_H_10_O_5_	150.0528
35	Isokaempferide	4.67	301.5982	257.5770 (100), 229.4959, 272.5718	C_16_H_12_O_6_	300.0633
14	8,8-DDPCM	8.19	339.8762	183.4248 (100), 299.6149, 197.4348	C_19_H_18_O_6_	342.1103
45	1-THONONE	8.44	311.175	183.5197 (100), 149.3188	C_15_H_20_O_7_	312.1209
46	7-Methoxyflavonol	8.49	266.6690	97.0762 (100), 245.4699	C_16_H_12_O_4_	268.0735

**Table 4 molecules-27-09012-t004:** List of some identified compounds abbreviated in Table 3.

Serial No.	Abbreviation	Compound Name
1	CGHTM, ME	Cyclopenta(c)pyran-4-carboxylic acid, 1-(beta-D-glucopyranosyloxy)-1,4a,5,6,7,7a-hexahydro-4a,7-dihydroxy-7-methyl-, methyl ester
2	8,8-DDPCM	(8,8-dimethyl-2,10-dioxo-9H-pyrano[2,3-f]chromen-9-yl) (Z)-2-methylbut-2-enoate
3	MDTT	Methyl (2E,4E,8E)-7,13-dihydroxy-4,8,12-trimethyltetradeca-2,4,8-trienoate
4	9,12,13,TriHODE	(10E,15E)-9,12,13-trihydroxyoctadeca-10,15-dienoic acid
5	EGCG	Epigallocatechin gallate
6	FA 18:1+3O	9,12,13-Trihydroxy-10-octadecenoic acid
7	FA 18:4+2O	(10E,12E,14E)-16-hydroxy-9-oxooctadeca-10,12,14-trienoic acid
8	9-HOA	9-Hydroxy-10E,12Z,15Z-octadecatrienoic acid
9	13-HODE	13-hydroxy-9,11-octadecadienoic acid
10	9,10-DiHOME	(12Z)-9,10-Dihydroxyoctadec-12-enoic acid
11	2-HCA	2-Hydroxycinnamic acid
12	1-THONONE	1-[2-methyl-6-[(2S,3R,4S,5S,6R)-3,4,5-trihydroxy-6-(hydroxymethyl)oxan-2-yl]oxyphenyl]ethanone

**Table 5 molecules-27-09012-t005:** IC_50_ values of *G. velutinum* extract and fractions for MCF-7 and PC-3 cell lines.

Entry No.	Sample Name	MCF-7 Cell Line IC_50_ (µg/mL)	PC-3 Cell Line IC_50_ (µg/mL)
1	Crude	431.78	89.02
2	*n*-Hexane fraction	523.63	325.87
3	Chloroform fraction	222.27	27.63
4	Ethyl acetate fraction	226.35	196.83
5	*n*-Butanol fraction	≤50% (Not applicable)	123.36
6	Aqueous fraction	522.41	36.95

## Data Availability

Upon request to the corresponding authors.

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
