# Peer review of "LC-MS/MS-Based Metabolomic Profiling of Constituents from Glochidion velutinum and Its Activity against Cancer Cell Lines"

_molecules, 2022, doi:10.3390/molecules27249012_

Round 1
Reviewer 1 Report
This can be an interesting article combining HPLC, MS/MS and cytotoxic results. However, it is quite short, the discussion about the results is not comprehensive.
I have some doubts regarding analytical methodology. Using an Orbitrap seems to provide High-Resolution Mass Spectrometric (HRMS) data – please explain in what mode was working this LC-MS equipment? What was possible to obtain? How many ions at once were analyzed, etc.
The results obtained for m/z are not so satisfactory – probably it would be better if the ions were grouped for monitoring and fragmentation in more than 1 analysis, so more time would be available for the ions.
We would expect the results for the obtained ion m/z with 4 digits after comma (or dot).
Table 1 – There are too many significant digits for these values. 4 digits seems enough here.
Fig. 1. If the m/z are not obtained in HRMS mode, the nominal masses are sufficient to be shown. This way the scheme will be more clear.
Table 3 – please write in what mode wad working the LC-MS – LRMS or HRMS?
The names in the 2-nd column are sometimes so long that the Table is too long. Please, prepare first another an introductory Table with a list – and abbreviations, so you could use abbreviations in the case of long names. Use lower size of the fonts in the Table 3
There are some formatting problems with the text – too much italic. There are no space lines between subsections.
Author Response
Dear esteemed Reviewer,
Please find our point-by-point rebuttal letter as a .pdf file, herein attached.
Again many thanks for your helpful comments and consideration.
Best,
The authors

Reviewer 2 Report
The paper ‘LC-MS/MS based metabolomic profiling of constituents from 2
Glochidion velutinum and its activity against prostate cancer 3
(PC-3) cell line’ describes the extraction and fractionation of leaves from Glochidion velutinum, followed by determination of metabolites and in vitro assessment of anticancer activity of these fractions in a PC-3 cell line.
The determination of metabolites using LCMS is fine, however, the work seems to be a bit incomplete in that the investigators reported only biological activity for fractions containing very many tentatively biologically active compounds. Thus, as a minimum, the following questions should be addressed before publication in Molecules.
· Did you find any unknown compounds during the profiling?
· The biological activities of most, if not all, described compounds have been published previously, a further discussion and references to earlier work are needed.
· There is no attempt to further fractionate and determine from where the main biological activity comes, didn’t you do this successfully?
· Only one cell line was used in the biological assessment. a couple more cancer cell types would be valuable, at least for the most active fraction(s).
Some small technicalities
· The result section is written in italics, why?
· Page 10 trehalose and citric acid have the same compound number in the table
· In Table 1 and 2, there are too many insignificant digits that are meaningful in terms of accuracy or precision e.g. Table 1, entry 1, 588.01 ± 2.97 should be 588 ± 3
Author Response

(The authors gave the same response as above.)

Round 2
Reviewer 1 Report
Tables 1 and 2 - Please, allign your data according to coma period)
In Abstract , there is no need to show exactly the data values – first of all , please remove the error data.
Lines 213-223 and Abstract the data in Abstract contain too many significant numbers. II propose a new Table containing all these data to compare
Figures 4 and 5 – labels on X-axis Please remove the digits after period
Line 361 – Please add an empty line before the formula
Table 3 – the column “Exact mass” – contains the theoretical mass? Please write 4 decimal digits
Author Response
Dear Editor: Molecules
We appreciate you for providing us an opportunity to revise the manuscript (Manuscript ID: Molecules-2046325) entitled “LC-MS/MS based metabolomic profiling of constituents from Glochidion velutinum and its activity against cancer cell lines" in the light of reviewer’s comments. Following is our point to point response to these comments. We have addressed them in earnest in the revised manuscript. All the changed/corrected parts have been highlighted in red. We are thankful for your efforts to improve our manuscript and are hopeful for positive decision on the revised manuscript
Reviewer # 1: Comments to Authors
- Tables 1 and 2; Please, align your data according to the coma period.
Response: Thank you very much for your comment. We have corrected and aligned Tables 1 and 2 in the revised manuscript according to your suggestion.
- 2. In the abstract, there is no need to show exactly the data values – first of all, please remove the error data.
Response: Thank you very much for your suggestion. We agree with your opinion that error data should be removed. We have removed the error data in the abstract according to the suggestion in the revised manuscript.
- Lines 213-223 and the Abstract contain too many significant numbers. Will propose a new Table containing all these data to compare.
Response: Thanks for your suggestion, we have adjusted the data numbers in lines 213-223 and Abstract. A new Table (Table 5) has been made according to the suggestion in the revised manuscript.
- Figure 4 and 5 – labels on the x-axis. Please, remove the digits after the period.
Response: Thank you very much for your comment. The digits after the period have been removed from Figures 4 and 5 as per your suggestion in the revised manuscript.
- Line 361, please add an empty line before the formula.
Response: Thanks for your suggestion, the empty line has been added before the formula in the revised manuscript.
- Table 3 – the column “Exact mass” – contains the theoretical mass? Please write 4 decimal digits.
Response: Thanks for your valuable suggestion. The column “Exact mass” in Table 3 has been modified and 4th decimal digit has been added accordingly in the revised manuscript.
Thank you very much for your insightful comments and suggestions. We sincerely believe that your comments and suggestions have significantly improved the quality of our revised manuscript for publication in Molecules Journal.
Sincerely,
Prof. Taous Khan, Ph.D.
Department of Pharmacy
COMSATS University Islamabad, Abbottabad Campus, Pakistan
taouskhan@cuiatd.edu.pk

Reviewer 2 Report
Overall the concerns raised by the reviewers have been addressed, even though this is a limited study it meets the requirements to be published in Molecules.
Author Response
Dear Editor: Molecules
We appreciate you for providing us an opportunity to revise the manuscript (Manuscript ID: Molecules-2046325) entitled “LC-MS/MS based metabolomic profiling of constituents from Glochidion velutinum and its activity against cancer cell lines" in the light of reviewer’s comments. Following is our point to point response to these comments. We have addressed them in earnest in the revised manuscript. All the changed/corrected parts have been highlighted in red. We are thankful for your efforts to improve our manuscript and are hopeful for positive decision on the revised manuscript.
Reviewer 2
Comments and Suggestions for Authors
- Overall the concerns raised by the reviewers have been addressed, even though this is a limited study it meets the requirements to be published in Molecules
Response: We are thankful to the reviewer for the favorable comments.
Sincerely,
Prof. Taous Khan, Ph.D.
Department of Pharmacy
COMSATS University Islamabad, Abbottabad Campus, Pakistan
taouskhan@cuiatd.edu.pk
